# Is Pediatric Intensive Care Trauma-Informed? A Review of Principles and Evidence

**DOI:** 10.3390/children9101575

**Published:** 2022-10-18

**Authors:** Lauren A. Demers, Naomi M. Wright, Avi J. Kopstick, Claire E. Niehaus, Trevor A. Hall, Cydni N. Williams, Andrew R. Riley

**Affiliations:** 1Pediatric Critical Care and Neurotrauma Recovery Program, Oregon Health & Science University, Portland, OR 97239, USA; 2Division of Pediatric Psychology, Department of Pediatrics, Oregon Health & Science University, Portland, OR 97239, USA; 3Department of Psychology, University of Denver, Denver, CO 80208, USA; 4Division of Pediatric Critical Care Medicine, Department of Pediatrics, Texas Tech University Health Science Center El Paso, El Paso, TX 97705, USA; 5Division of Psychology and Psychiatry, University of Louisville School of Medicine, Louisville, KY 40202, USA; 6Division of Pediatric Critical Care, Department of Pediatrics, Oregon Health & Science University, Portland, OR 97239, USA

**Keywords:** pediatric intensive care unit, pediatric critical care, trauma-informed care, medical trauma, pediatric medical traumatic stress

## Abstract

Pediatric critical illness and injury, along with the experience of recovering from critical illness are among the most potentially traumatic experiences for children and their families. Additionally, children often come to the Pediatric Intensive Care Unit (PICU) with pre-existing trauma that may sensitize them to PICU-related distress. Trauma-informed care (TIC) in the PICU, while under-examined, has the potential to enhance quality of care, mitigate trauma-related symptoms, encourage positive coping, and provide anticipatory guidance for the recovery process. This narrative review paper first describes the need for TIC in the PICU and then introduces the principles of TIC as outlined by the American Academy of Pediatrics: awareness, readiness, detection and assessment, management, and integration. Current clinical practices within PICU settings are reviewed according to each TIC principle. Discussion about opportunities for further development of TIC programs to improve patient care and advance knowledge is also included.

## 1. Introduction

### 1.1. Pediatric Medical Traumatic Stress Following Pediatric Critical Care

Each year, over 300,000 children sustain life-threatening injury or illness and require the highest level of care through hospitalization in pediatric intensive care units (PICUs) [1]. These children and their families experience difficult and potentially traumatic events including illnesses and injuries that are often unexpected and fear-provoking. Hospitalization brings additional stress and challenges, due to emerging or worsening physical or emotional symptoms, unfamiliar and/or invasive medical procedures, and difficult decisions about high-risk treatments, end-of-life, and/or palliative care [2]. Since advances in technology such as extracorporeal life support, noninvasive ventilation, and noninvasive diagnostic and monitoring techniques have improved survival rates, recent focus on improving PICU management has shifted to addressing increased morbidity rates [3] and impacts on quality of life, including psychological outcomes.

The assortment of physical, cognitive, emotional, and social challenges of PICU survivors are termed post-intensive care syndrome (PICS) [4]. This includes posttraumatic stress symptoms (PTSS), and specifically pediatric medical traumatic stress (PMTS), which are common emotional, behavioral, and physiological sequelae in children and their families after severe medical illness, injury, and life-saving treatment [5]. Research on risk factors for negative functional outcomes after PICU hospitalization has uncovered the role of injury-related factors, including increased number of invasive procedures and interventions, type of illness (e.g., chronic illness or physical trauma), and increased benzodiazepine and narcotic administration, as well as demographic factors such as younger child age, lower family socioeconomic status, or critical access hospitalization [6,7,8]. PMTS may manifest as stress-related disorders, such as acute stress disorder and post-traumatic stress disorder (PTSD), or as symptoms of arousal, avoidance, and re-experiencing that do not rise to a pathological level [9]. These psychological symptoms in children are associated with negative health outcomes [10,11], including poorer treatment adherence, lower health-related quality of life, and impaired physical and emotional health [12,13]. Similarly, parental or caregiver distress and PMTS symptoms predict poorer child outcomes in general [14] and specifically in post-PICU populations [15,16,17].

Though there has been less empirical attention on subjective factors that confer risk for PMTS, the primary role of the patient and family’s perception and experience of illness has been integrated into a comprehensive model of PMTS, the Integrative Trajectory Model of PMTS [18,19]. According to this empirically supported model, PMTS severity over time is driven largely by family members’ “subjective appraisal and responses” during the course of the medical events, from time of injury to the time past the end of active medical treatment. Children often remember the incident that led to admission and some distressing events in the PICU including treatment-induced hallucinations and waking up and not knowing where their parents were [20]. Parents have identified the most stressful aspects of the PICU as changes to family functioning and the parenting role, uncertainty of the child’s outcome, sights and sounds, and child appearance and perceived acuity [21,22]. An emergent body of work has demonstrated that children’s and parents’ subjective experience of the illness and hospitalization consistently predicts PMTS more than objective elements of the illness, such as severity and length of hospitalization [20,23,24,25].

### 1.2. Premorbid Trauma and Pediatric Critical Care

While much has been learned about the traumatic nature of severe injuries/illnesses, critical care, and PICU hospitalization overall, little attention has been paid to premorbid trauma and how it may interface with the PICU environment, despite the prevalence of such trauma. Nearly two-thirds of youth are exposed to at least one significant adverse event or ongoing experience, such as emotional, sexual, or physical abuse, and household substance abuse or mental illness [26]. Moreover, rates of childhood traumatic experiences have been found to be elevated in critically ill children [27]. Recent work examining trauma symptom trajectories following PICU hospitalization demonstrated that pre-existing internalizing child behavior predicts ongoing PMTS [16]. Patients with a history of medical trauma or developmental differences are also much more likely to become dysregulated during a very stressful treatment course [28]. Consideration of life experience is prudent, as it has been postulated that prior adverse life events sensitize children to psychological distress in the PICU [16,27].

### 1.3. Trauma-Informed Care

In response to mounting evidence about the prevalence and impact of traumatic experiences, a trauma-informed approach to practicing medicine first developed in the 1970s [29], took hold in the late 1990s [30], and has been further advocated for within child health systems in recent years [31]. Trauma-Informed Care (TIC) refers broadly to physical health, behavioral health, or social services that are provided in a way that acknowledges the prevalence of traumatic experiences among patients, their family members, providers, and staff; accommodates the needs and vulnerabilities of trauma survivors; and minimizes the likelihood that re-traumatization occurs within care systems [32].

From inception, TIC was grounded in trauma theory and consisted of specific actions and principles that would characterize TIC in varied service settings [32,33,34]. In recent years, core TIC principles have been applied to healthcare settings [35], including those serving pediatric populations [31,36,37]. Forkey and colleagues’ [31] American Academy of Pediatrics Practice Guideline organizes TIC clinical strategies into five categories: awareness, readiness, detection and assessment, management, and integration. *Awareness* refers to the need for providers to have a strong working knowledge of the science of trauma (e.g., what are potentially traumatic events; what is the range of physical, cognitive, emotional, and behavioral reactions to traumatic stress; how might families’ culture or context affect traumatic stress reactions; who is at greater risk for adverse reactions to trauma). *Readiness* involves being prepared to implement strategies to promote resilience in both child and caregiver. *Detection and assessment* refer to strategies for universally identifying traumatic stress reactions and potentially traumatic premorbid experiences. *Management* refers to addressing effects of trauma in the short-term so medical care can happen, as well as longer-term intervention for trauma sequelae. *Integration* refers to TIC becoming standard practice within medical settings and extending to other settings where patients receive aftercare.

### 1.4. Aims

The current paper aims to highlight and evaluate existing evidence pertinent to TIC elements in the PICU and to provide recommendations for enhancing TIC in the PICU. This review focuses on the clinical principles and practices that can be implemented at the ground level with patients and families; however, it is critical to acknowledge the necessity of implementing TIC clinical strategies with adequate organizational support.

## 2. Methods

We conducted a traditional or narrative review. A PubMed, WorldCat, and Google Scholar database search was conducted using various combinations of the terms: trauma informed care, pediatric intensive care unit, critical care, medical trauma, intervention, secondary traumatic stress, burnout, and pediatric or children. For select articles, references cited were reviewed and the “cited by” option was used as an additional search method. These techniques have been referred to as “snowballing” [38]. Literature on TIC or the ICU, individually, was reviewed as well for supplemental purposes. Articles referenced for the purpose of this review were published between 1988 and August 2022. The authors (LD, NW, AK, and CN) reviewed eligible articles and excluded articles with no relevance, case reports, and non-English language studies.

## 3. Results

### 3.1. Awareness

Despite growing recognition of trauma and its impacts by the general public and popular media [39], integration of these concepts into medical settings is relatively nascent. A 2017 Council of Academy Family Medicine Educational Research Alliance (CERA) survey of program directors indicated fewer than a third of family medicine programs included TIC in their curricula [40]. As Forkey et al. [36] identified, there is a need for further awareness and understanding of key concepts and terminologies such as social determinants of health, toxic stress, adverse childhood experiences (ACEs), complex childhood trauma, developmental trauma disorder, and secondary traumatic stress (STS). Trauma-informed awareness additionally requires appreciation of how cultural and material contexts affect persons’ experience of and reaction to traumatic events. For example, traumatic distress may present along a continuum from subdued, calm, or numbed to hyper-reactive or demonstratively upset, and differences in the phenomenology of trauma responses have been observed across cultures [41]. Providers must therefore reach beyond their own culturally bound conceptions of what “appropriate” distress looks like to better interpret the behavior of patients and families in acutely stressful situations. Further, variations in families’ trust of medical systems and personnel can be affected by cultural and trauma-related factors [42]. Persons of different cultural backgrounds, educational levels, and other disadvantaged groups are likely to have differing experiences and interactions with healthcare systems, including those that are perceived as discriminatory. Perceived discrimination has important implications for health outcomes [43], so provider awareness of these issues is essential to promoting wellness.

Few studies have attempted to assess or influence TIC awareness among PICU staff or pediatric hospital workers more broadly. A retrospective chart review of PICU providers concluded that “trauma” still mostly connotes blunt trauma, and few patients ever receive psychosocial diagnoses or interventions to mitigate lasting effects of trauma [44]. Similarly, a survey of adult ICU providers about TIC practices focused only on the aftermath of blunt force trauma (in the form of PTSS), as opposed to pre-existing factors [45]. Beyond intensive care, one survey study of tertiary children’s hospital workers in Australia found that, overall, TIC-awareness was low-to-moderate, with hospital administrators considering themselves to be the least educated about trauma and psychosocial care [46]. The authors noted the potential of formal training to improve individuals’ knowledge, confidence, and use of TIC. Finally, an implementation study found that surgical residents reported low understanding of the physiology of the fear response, the neurobiology of trauma, aspects of the brain involved in the process of fear conditioning, and the connection between fear, trauma, and aggression. This awareness improved only slightly in response to a trauma-focused education session [47].

### 3.2. Readiness

Whereas awareness refers to knowledge of trauma and its impacts, readiness pertains to an understanding of processes and strategies that protect against the ill-effects of trauma. Little research has attempted to assess TIC readiness in the PICU, specifically, but a few studies have assessed providers’ knowledge of and attitudes toward psychosocial interventions more broadly. One modified Delphi-method study of PICU and pediatric palliative care providers found that extended family support, caregivers being in a committed relationship, and caregiver optimism were viewed as consensus protective factors [48]. Another study used qualitative methods to capture the perspectives of emergency, intensive care, and acute care providers on psychosocial interventions following severe TBI [49]. The researchers identified an overall desire to implement family-centered and trauma-informed practices, specifically by enhancing communication and collaboration with family members and better coordinating multi-disciplinary care.

It should be noted that providers also report barriers to readiness for TIC. Front-line professionals report that TIC increases expected workload and performance targets in already strained roles without the capacity to properly capture trust, safety or empowerment outcomes for their patients with complex trauma [50]. In an urban tertiary academic medical center, blunt trauma responders similarly reported time constraints, need of training, confusing information and evidence on trauma-informed practices, and worry about further upsetting or re-traumatizing patients, as factors preventing them from delivering TIC [45]. These barriers highlight a need for systemic improvements in the form of policy changes that support a TIC competent workforce\to achieve true TIC readiness in the PICU and beyond [31].

### 3.3. Detection and Assessment

Although children and families admitted to a PICU are at significantly elevated risk for developing traumatic stress and other mental health symptoms [51,52,53], there is no universal standard for how to best screen for trauma-related distress among hospitalized children or their caregivers. Indeed, among U.S. Level 1 Trauma centers, only 20% reported specialized trauma symptom screening and intervention services for children and families [54]. Assessment tools used during admission can help identify current hospitalization-related stressors, as well as those patients and families who are at greatest risk for developing post-PICU distress [52]. Much of what is known about trauma assessment in PICU settings is gleaned from research protocols, which did not evaluate the feasibility of integrating universal or targeted assessment as part of the task flow of a clinical team. However, a small number of PICU teams have reported descriptions of integrated trauma assessment protocols. Three PICU teams attempted a universal screening approach, though all missed some families due to limitations during roll-out, families declining to participate, or patients discharging before assessment could be completed. The first PICU team implemented a standardized family stress screening tool and response protocol [55]. The assessment was led by a provider with “expertise in assessing family stress/coping and ability to address family needs” [55]. Compared with families admitted before the protocol rollout, screening was associated with increased parent satisfaction and decreased security calls for distressed families. The other two integrated screening approaches were implemented by social workers and nurses who received a brief training [56] and a psychologist [57]. Of the three approaches, only Samuel and colleagues [57] assessed acceptability; 85% of caregivers reported no distress. In sum, there is preliminary support for universal screening by trained staff of children and/or caregivers in the PICU for trauma exposure, traumatic stress, and broad mental health symptoms.

#### 3.3.1. Evidence-Based Assessment Tools: Child and Family Distress

As summarized in Table 1, a number of evidence-based assessment tools have been used to assess trauma-related distress of children and families in PICU settings. These include the feasibility studies of universal screening described in the previous section, as well as assessment tools that were incidentally included to measure a research construct. Available tools vary with regard to the constructs assessed, respondent, and duration of administration. Most studied tools assess frequency of acute or traumatic stress symptoms. One alternative approach is to assess risk factors for subsequent PTSD, which are gleaned from child and caregiver reports, as well as chart review [58].

#### 3.3.2. Evidence-Based Assessment Tools: Provider Distress

Up to half of all PICU providers report significant STS (i.e., indirect exposure to others’ traumatic experiences through hearing about or caring for traumatized persons), contributing to provider turnover and poor provider well-being [59]. While there has been little investigation of the detection and assessment of trauma-related distress of providers in PICUs, the phenomenon has been explored in adult ICUs (for a review, see van Mol et al. [60]). In addition to STS, other relevant constructs can also affect providers, including burnout (i.e., impairment produced by occupational stress) and moral distress (i.e., being overwhelmed by powerlessness to do what is believed to be right; sometimes called “compassion fatigue”) [61]. Among PICU-specific studies, Flanders and colleagues [62] sought to implement and assess the real-world feasibility of a tool that assesses provider wellbeing. They used the Professional Quality of Life measure [63] to assess compassion satisfaction and fatigue levels, which comprise secondary traumatic stress and burnout. The 30-item measure uses a 5-point Likert response scale and can be completed quickly. Another PICU-specific study [64] trialed the use of self-report measures of wellbeing for PICU providers, using the Moral Distress Scale-Revised questionnaire [65], Trauma Screening Questionnaire [66], and abbreviated Maslach Burnout Inventory [67].

**Table 1 children-09-01575-t001:** Screening Tools Studied with Children and Caregivers in PICU.

Measure Citation	Construct	Reporter	Length	Question/Response Type	Timing of Administration	Study in the PICU	Reliability & Validity
Screening Tool for Early Predictors of PTSD (STEPP; Winston et al., 2003) [58]	Risk factors for child developing subsequent PTSD	Child (age 8–17), Caregiver, and Provider/Chart Review	12-items (4 per reporting source)	Yes/No	In PICU, *M* = 2.3 days after admission	Kassam-Adams et al. (2004) [68]	Moderate test re-test reliability of facial events (*k* = 0.60), poor reliability for emotional states (*k* = 0.25) and substantial for children getting a positive screen (*k =* 0.86); validity not reported
Acute Stress Checklist for Children (Kassam-Adams et al., 2006) [69]	Child acute stress symptoms	Child (age 8–17)	19 items	3-point Likert	“As soon as child was well enough to do so” during admission	Nelson et al. (2019) [70]	Expert panel rated highly for validity and highly correlated with similar measures (*r* = 0.77); (α = 0.85; 19 item version) and (α = 0.86; 26 item version). 1 week test–retest reliability (*r* = 0.76; 19 item) and (*r* = 0.78; 26 item)
Child PTSD Symptom Scale (Foa, Johnson, Feeny, & Treadwell, 2001) [71]	Child traumatic stress symptoms	Child (age 8–17)	17 items	4-point Likert	In PICU, *M* = 2.3 days after admission	Ewing-Cobbs et al., 2017 [72]	Good internal consistency (α = 0.89) for total score and convergent validity with similar measures (*r* = 0.80); moderate re-test reliability (*k =* 0.55)
Posttraumatic Adjustment Screen, modified to focus on child’s reason for admission (O’Donnel et al., 2008) [73]	Risk of caregiver PTSD and depression after traumatic events	Caregiver	10 items	5-point Likert	Within 48 h of admission [74]; At time of PICU discharge [57]	Samuel et al. (2015) [57]	Using ROC curve analyses, demonstrated adequate sensitivity (0.82), specificity (0.84), and correctly classified the PTSD outcome of 84% of participants at 12 months
Impact of Event Scale, Revised (Weiss & Marmar, 1997) [75]	Caregiver traumatic stress symptoms	Caregiver	22 items	5-point Likert	At time of PICU discharge [57]; 1–8 days post PICU discharge [76]	Samuel et al. (2015) [57]; Wawer et al. (2020) [76]	Subscale internal consistency was adequate (α = 0.79–0.92); concurrent validity with measures of PTSD, depression, and anxiety (*p*’s < 0.05) [77].
Stanford Acute Stress Reaction Questionnaire (Cardeña et al., 2000) [78]	Caregiver acute stress symptoms	Caregiver	30 items	Semi- structured interview	“As soon as child was well enough to do so” during admission	Nelson et al. (2019) [70]	Good internal consistency (α = 0.90) and convergent validity with measures of PTSD (ρ = 0.79).
Parental Stressor Scale: Pediatric Intensive Care Unit, Revised (Carter & Miles, 1989; revised version Alzawad et al., 2021) [79,80]	Impact of PICU environmental stressors on caregivers	Caregiver	41 items	6-point Likert	Varies; as little as 48 h after admission [81]	See Alzawad et al. (2021) [80] for review	Moderate to strong subscale test–retest reliability, *k* = 0.58–0.92 [82]; acceptable to good internal consistencies (α = 0.68–0.91) [80]; good subscale convergent validity with measure of state-trait anxiety (*r* = 0.29–0.42, *p* < 0.0001) [79].
Family Stress Screening Tool (Liaw et al., 2019) [55]	Family system distress	Caregiver or Provider ^1^	1 item	Distress rating scale (thermometer)	24–48 h after admission	Liaw et al. (2019) [55]	Reliability and validity not reported.

^1^ For families in high-stress situations (e.g., too distressed post-admission), the FSST can be used to determine a perceived stress score from the provider perspective.

### 3.4. Management

There are no published standards of practice or guidelines with respect to managing historical trauma in the PICU. Further, the extant literature is scant in terms of description and review of TIC management systems practiced in this setting. Researchers at a leading children’s hospital detailed the vast benefits of implementing TIC within pediatric healthcare networks broadly and provided a framework for doing so [11]. The framework includes three core elements: (1) Reduce distress (minimize potentially traumatic aspects of medical care; address patient pain, fear, and grief); (2) promote emotional support (provide reassurance and emotional support; encourage positive coping); and (3) attend to the family (promote family presence and participation in care; address family worries; give families accurate information, reassurance, and shared decision-making power).

#### 3.4.1. Reduce Distress

In terms of minimizing and addressing patient distress in the PICU, recent research indicates inconsistent application of TIC principles. The Pediatric PAUSE (Pain and Privacy, Anxiety and IV Access, Urinary catheter/rectal exam/genital exam, Support from family or staff, Explain to patient/Engage with PICU team) Protocol is the most comprehensive TIC approach described in the literature [83]. The protocol is designed to be performed between primary and secondary medical trauma surveys with the goal of attenuating patient pain and anxiety. Trauma, emergency, and pediatric intensive care departments, work collaboratively to determine whether further pain interventions are indicated, which personnel are needed at bedside while the secondary survey is performed, how to reduce patient fears with age-appropriate language, whether genital and rectal exams are indicated, whether urinary catheter is necessary, which family member or team member can be present at the head of the patient’s bed to comfort the patient during exam, and how to communicate with the patient in an appropriate manner about upcoming procedures and interventions. The protocol takes about 5 min and has been shown to not delay first imaging in pediatric trauma patients.

Several studies have examined the practices PICU providers use to mitigate patient pain and fear, though grief has not yet been explored. These strategies include mental imagery to distract from physical pain [84] and ICU diaries to reduce patient and family distress [85,86,87]. Unfortunately, high PICU workload and limited provider familiarity with these practices have been identified as barriers to implementing such strategies [84,86].

There is minimal empirical research on practices designed to reduce potentially traumatic aspects of medical care or common trauma triggers during PICU hospitalization; however, three recent systematic reviews focused on interventions targeting PMTS following critical injury or illness [88,89,90]. Baker and Gledhill identified 3 randomized controlled trials and 3 feasibility studies aimed at reducing psychiatric morbidity in patients and their caregivers after PICU discharge [88]. Another systematic review included 5 of the 6 studies as Baker and Gledhill’s review, as well as an exploratory cross-sectional study on mothers’ views on the potential value of a follow-up appointment [24]. The authors of both reviews emphasized the benefit of parental support after discharge, whether by appointment, telephone check-in, or provision of written psychoeducational materials. However, they cautioned that the current evidence is preliminary and insufficient to support any intervention in isolation [88,89]. Finally, Christian-Brandt and colleagues reviewed 16 intervention studies that targeted PMTS of patients affected by childhood cancer or acute medical trauma [90]. They determined that the interventions were generally consistent with best practices for treatment of childhood trauma, including caregiver involvement and utilization of cognitive behavioral therapy principles. They noted that not all interventions carefully considered the developmental stage of the child. Further, they found that although all studies reported improvements in PMTS, the most methodologically rigorous studies found limited intervention effects relative to control groups. They concluded that the most promising interventions were online, self-guided, or time-limited. Across all three reviews, the authors highlighted the general paucity of research on intervention studies for this population, despite great need. Studies tend to focus on families affected by pediatric traumatic brain injury in particular, though outcomes are mixed [91,92,93,94,95]. Of note, TIC is distinguished from these types of trauma-specific interventions designed to directly ameliorate the effects of past trauma [96].

#### 3.4.2. Promote Emotional Support

There is also minimal research on the nature and pervasiveness of emotional support within the PICU setting. One recent study examined the use of TIC and psychosocial care in a pediatric hospital. Seventeen staff members, including 5 PICU providers, were interviewed about how they view psychosocial care and when they use it [46]. Only half of interviewed staff reported that they offer psychosocial care such as psychological support and relationship building to all patients, and the majority instead rely on a noticeable trigger (e.g., crying, stated fear of needles). The most commonly reported barrier to providing psychosocial care was lack of time due to competing demands (n = 13). An alarmingly high number of providers also cited barriers that may actually be indicative of patient trauma responses (i.e., refusing care, aggression; n = 11).

#### 3.4.3. Attend to the Family

Researchers have called on PICU staff and management to build upon and strengthen patients’ naturally occurring social support systems [56]. Most PICUs are set up to encourage and support family involvement by having favorable visitation policies and dedicated, comfortable spaces for families [97], though restrictions increased during the COVID-19 pandemic. The increasingly common practice of including family members in symptom management planning during interdisciplinary rounds has been found to reduce costs and length of hospitalization, enhance collaboration between team members, and increase family medical understanding, psychosocial functioning, and confidence in the health care team [98]. However, many units do not have standard protocols for involving parents in their children’s care. In an international survey of PICUs, only 23% of centers reported providing families with formal instruction for participating in their child’s care [99]. It has been recommended that providers create opportunities for family involvement in patient care by teaching family members to handle mouth care and range-of-motion exercises [97]. Parental touch and talk have also been promoted as methods of comfort and distraction during procedures, as they have been found to relate to shorter recovery (i.e., time to return to baseline heart rate) after invasive procedures [100]. A similar protocol that involves nightly parental soothing through touch, reading, and music has been found to be feasible in the PICU and wards, judged to be calming by nurses and parents, and potentially anxiety-reducing for patients and parents [101].

Existing evidence shows a clear need to enhance family-provider partnership in PICU settings. For instance, in a thematic analysis of questionnaires completed by 70 parents of children cared for at four Swedish PICUs, results revealed that they viewed partnership as lacking, particularly in terms of person-centered communication and decision-making about care and treatment [102]. Need for more clear communication between physicians and patients has also been identified in research about shared understanding regarding prognostic conversations [103]. Some hospital systems have formalized partnership programs to address these needs. Inclusion of a trained navigator who provides emotional, communication, decision-making, information, and transition support to the patient’s healthcare team has garnered preliminary support, though the research has been underpowered [104]. A study on a mother-nurse partnership program for parents of infants in a pediatric cardiac ICU also yielded promising results [105]. The program, geared to promote information sharing, negotiation, and participation in care, was associated with higher parental PICU satisfaction, self-efficacy, and perceived partnership, alongside lower anxiety upon transfer to the ward. Overall, current partnership and shared-decision making practices are often insufficient and lack evidence-based communication strategies to empower families to be involved in their children’s care.

Several pilot randomized controlled trials during and/or shortly following PICU hospitalization have included interventions designed to reduce parental anxiety, increase knowledge, and impart skills to mitigate PICS symptoms. The largest and most comprehensive of these studies focused on 2- to 7-year-old children. The intervention involved providing psychoeducation material, anticipatory guidance on the hospitalization and recovery processes, child coping suggestions and interactive exercises (i.e., parent–child activity workbooks, storybooks, puppets), and a check-in telephone call several days after hospital discharge [106]. Mothers who received the intervention showed improved emotional functioning, greater involvement in their children’s physical and emotional care, and rated their children as exhibiting fewer withdrawal and behavioral symptoms post-discharge. A recent review reported similar results of five trials designed to support parents of infants hospitalized for congenital anomalies [107]. Interventions included psychoeducation, parenting skills training, parent-infant interaction and bonding, early pediatric palliative care, and family-centered nursing. Though the quality of evidence was considered low, most trials reduced maternal anxiety and garnered mixed results for material depression. In contrast, another pilot randomized controlled trial that focused on child and adolescent outcomes did not yield significant results [56]. This pilot study found an intervention comprising family interviews, educational materials, and post-discharge care coordination did not produce significant changes in child mental health symptoms or quality of life. Finally, a recent study comparing parental learning outcomes across brochures, scripted conversations, and 3-min videos about post-intensive care syndrome found that all three methods were useful and minimally disruptive to nurse workflow, though no information was collected on adoption of coping skills and parenting techniques, nor psychological outcomes [108]. Further research should aim to elucidate the differences in outcomes between these studies, and whether they are due to the nature of the intervention programs themselves, the age and risk-level of patients, the selection of measures and informants, and/or the timing of measurements.

### 3.5. Integration across Timelines and Settings

Forkey and colleagues [36] describe integration as TIC becoming standard of care through application of trauma-informed principles into policies, procedures, and patient care including staff training, integrated psychological services for patients and families, and support for staff. As described above, the elevated rate of trauma exposure and post-traumatic symptoms in the PICU warrants efforts to embed TIC in all aspects of the patient experience during their PICU encounter, and throughout follow-up care.

One option for supporting TIC integration in the PICU is the co-location of behavioral health professionals. This approach allows for the provision of evidence-based trauma interventions for families and children in the acute aftermath of a traumatic event, as well as after discharge (e.g., follow-up clinic). Several models of PICU-integrated behavioral health services have been described. One group implemented integrated behavioral health services in the medical/surgical and cardiac intensive care units of a pediatric hospital [109]. They found the service was most commonly provided for end-of-life care, children with longer hospitalizations, unexpected critical care admissions, accidental trauma, and new onset chronic illness. Most referrals (71%) were related to behavior management concerns (e.g., medication or procedure adherence, coping with hospitalization, end of life support), with 20% related to diagnostic clarity and the remaining 9% related to mental health discharge planning [109]. A second group has developed and reported on the Proactive Pediatric Psychology Consult Liaison model (PPPCL) [110]. This service, embedded in the PICU and general pediatric medical units, is staffed by a psychologist or psychology resident five days per week. PPPCL clinicians attend rounds, participate in interdisciplinary care conferences, and provide direct care to families as indicated. Common reasons for referral to their program include providing acute stress interventions after traumatic events, de-escalation of patients and families, and providing coping support for grief and hospital experiences. Patients with significant mental health symptoms who express interest in longer-term supports are referred to outpatient behavioral health post-discharge.

Beyond the PICU encounter period, implementation of TIC for PICU patients must also extend throughout patients’ transfer and discharge from the PICU. While little research has examined how TIC can be implemented during step-downs in levels of care, one approach, which included providing written and verbal information about what to expect before the transition, was found to be effective at preparing families for their inevitable stepping down from the PICU [111]. Most children admitted to a PICU will discharge within one month [112], which is less than the required time threshold for diagnosing persistent Acute Stress Disorder symptoms as PTSD. This timeline therefore suggests that post-discharge behavioral health follow-up for PICU patients is critical. While follow-up clinics for adult ICU survivors have existed for decades [113], similar programs are rare for PICU survivors [114], with the few existing programs embedded within larger children’s hospital systems. Of the PICU follow-up clinics that have been described in the literature, only 12% included a psychologist team member and one in three programs were physician only [114]. Most mainly provided screening for psychological morbidity and behavioral health care services for children after their PICU stay. While the vast majority of PICU survivors and families do not receive this coordinated care, many attend follow-ups with their pediatrician or in a specialty clinic [114,115]. Increasingly, pediatric primary care and specialty clinic settings are incorporating integrated behavioral health providers who could conduct universal behavioral health screenings with patients during post-discharge appointments, including screening for traumatic distress with patients and caregivers [116,117]. Taken together, both integrated and ongoing post-discharge behavioral health services should be considered essential to caring for a population at high risk for posttraumatic stress.

Integration also entails strategic attempts to universally mitigate providers’ STS, burnout, and moral distress as well as mechanisms for identifying and addressing elevated distress when it occurs. Despite longstanding awareness of the problem of PICU/ICU provider distress [118], few interventions have been developed or evaluated. Several pilot studies described potential strategies, including a psychoeducational webinar for providers, a book club, a retreat, available counseling, and regular planned debriefings integrated into standard workflow [119,120]. While these interventions appeared to be well-received based on provider report, no quantitative evaluation of STS or related distress was conducted. The one program to our knowledge that was evaluated for effectiveness at reducing provider moral distress entailed a once-weekly facilitated discussion focused on addressing care goals and ethical issues [121]. While provider moral distress did not significantly decrease after implementation of the intervention overall, some individual items elicited less moral distress after the program (e.g., “initiating extensive life-saving actions when I think they only prolong death”).

## 4. Discussion

The extant literature supports providing TIC universally. The following examples, illustrated in Figure 1, demonstrate how TIC clinical strategies can alter care for children and families in a PICU setting. Consider patients and families with pre-existing historical or ongoing trauma who are confronted with the additional violation of physical integrity and loss of autonomy inherent to critical illness or injury (e.g., physical examinations, restraints, medical procedures, and treatments). They may exhibit a series of adaptive responses to trauma: a hyperarousal or a dissociative state [9]. If providers are not mindful of potential pre-existing trauma, the core symptoms of PTSD, and PTSS more broadly, these actions may be misinterpreted. Symptoms of hyperarousal and agitation resulting from the trauma response could be misinterpreted as hyperactive delirium (another common complication in the PICU) or lead to additional use of medications for sedation that can exacerbate risk for development of delirium and delusional memories exacerbating medical stress. Alternatively, catatonia as a trauma response can be misinterpreted as hypo-active delirium or encephalopathy for which treatments would differ. Likewise, in family members, hyperarousal trauma reactions may be misinterpreted as inappropriate nervous excitement, defensiveness, and being a “difficult parent,” while dissociative reactions may be misinterpreted as being rude, standoffish, or not invested in the patient’s recovery [97]. Such families are less likely to receive patient/family-centered care and thereafter be less likely to be actively involved in the medical decision-making and recovery processes. In turn, patients and families may further withdraw from the medical system and become further traumatized.

In contrast, trauma-informed providers will be knowledgeable and will consistently provide psychosocial support and universal trauma precautions, such as by describing upcoming procedures in accordance with the family’s level of health literacy and communication preferences, inviting questions about care and prognosis, providing reassurance, and offering choices when possible. This will allow for families to be more involved in shared medical decision-making and follow-up care. Trauma informed-providers will also assess for traumatic stress reactions with evidence-based tools and clinical knowledge, allowing them to recognize agitation, restlessness, or rapid changes in mood during or when discussing procedures and examinations, as trauma-related symptoms. This understanding will allow for more empathetic and family-centered medical caregiving, and more effective care. Further, TIC affords opportunities to minimize exposure to triggers and enhance individual and family coping during and after hospitalization. In turn, patients and families may be more cooperative during procedures and recover faster, such that use of additional emergency services is reduced.

While this article has focused on integrating TIC into clinical practice, it is critical to have organizational structures that promote TIC through policy and allocated resources. Indeed, individual clinicians implementing TIC without leadership support is likely unsustainable and may actually increase secondary traumatic stress (STS) for providers. For example, if new services, such as integrated behavioral health in PICU settings, are implemented, creative leadership will be necessary to identify how such services can be billed to insurance. Organizational, leadership, and policy action steps have been outlined elsewhere [31,122]. Below, we describe key clinical recommendations about enhancing TIC in the PICU, based on the extant literature and informed by our clinical experience.

### 4.1. Awareness and Readiness Recommendations

Recent advances in TIC medical education show promise for preparing future trauma-informed providers. Examples include the trauma-informed medical education (TIME) framework developed at Harvard Medical School [123], models that incorporate TIC into existing medical school courses [124,125], and didactic and reflective practices built into resident training [126]. Specific to PICU physicians, TIC training that emphasizes the nuances of TIC in the PICU could be incorporated into PICU fellowship training.

Excellent and accessible resources also exist for current providers interested in learning more about the impacts of trauma (e.g., *What Happened to you?: Conversations on Trauma, Resilience, and Healing* by Oprah Winfrey and Bruce D. Perry, *The Body Keeps the Score: Brain, Mind, and Body in the Healing of Trauma* by Bessel van der Kolk) and implementing related healthcare practices, such as cultural humility [127,128,129]. Beyond the basic principles of cultural and contextual awareness and humility, we suggest providers develop a curiosity towards how trauma and culture interact to influence distress expression and experiences within healthcare systems.

### 4.2. Detection and Assessment Recommendations

Recommendations regarding screening for exposure to traumatic events and trauma-related distress come with caveats. Anecdotally, some PICU providers are wary of facilitating screening, typically citing fear of re-traumatization or lack of training in how to respond. Similar provider reluctance has been documented in other settings [130]. To address these concerns, those conducting trauma-specific assessments should have adequate training and resources to address identified needs. Despite challenges, evidence suggests screening and intervention is more effective when conducted earlier [131], so we recommend universal screening for acute stress and broader mental health symptoms for children and families in the PICU. Depending on available resources, screening results can be used to triage behavioral health services to patients most likely to develop PTSD or other chronic sequelae.

Providers may also question whether to limit assessment to the traumatic event that brought the child into the hospital or to assess for lifetime exposure to traumatic events. We recommend focusing on current trauma symptoms (e.g., acute stress for children and caregivers, and STS for providers) whether they are stemming from a historical event or a current event. This allows providers to meet patients’ current needs and conduct assessment with the purpose of guiding an immediate response as opposed to opening up painful memories without supporting treatment. For example, recent controversy over ACEs screening in primary care clinics highlights the potential harm of asking people about traumatic events with insufficient follow up to address trauma-related distress [132]. Similarly, we recommend using validated tools as opposed to providers interviewing children in detail about their traumatic experiences, which could be retraumatizing, especially if providers are not well trained in trauma-focused work. Despite the potential challenges of assessing patient and caregiver distress, better understanding families’ needs can help providers reduce stressors during hospitalization and tailor referral to resources after discharge. Further, as Liaw and colleagues observed, regardless of the assessment tool used, simply asking families about how they are doing has the potential to alleviate stress, identify easily addressed concerns, and foster the connection between the family and PICU team [55].

With respect to assessing provider distress, it should be stressed that, regardless of the specific tool chosen to assess provider wellbeing, the assessment procedures should be considered carefully to ensure provider autonomy and confidentiality. Assessment results should not be used punitively (e.g., restricting hours of a provider who screens positive). As with patient-focused assessment, there should be a clear follow-up plan for providers who indicate significant distress.

### 4.3. Management Recommendations

Reducing potentially traumatic and distressing aspects of medical care will look different for each patient and family, depending on their historical and ongoing trauma, individual and cultural identity, and medical presentation. However, all patients deserve universal trauma precautions, such as describing what will happen during each clinical encounter or procedure, inviting questions, and offering choices [35]. We also recommend best practices be applied within the PICU setting to enhance psychological safety: consistent psychosocial support, behavioral health support, and sufficient pain management and/or palliative care [97]. Given the important role of parents as attachment figures and sources of comfort, it is essential to effectively communicate with and involve them in their children’s care. The Pediatric PAUSE protocol described above is a promising TIC intervention designed to mitigate traumatization, reduce distress, and involve families in care and has been proven feasible [83].

Small but meaningful steps can be taken to involve parents in both physical care including mouth care and range of motion exercises, and emotional care such as by providing comforting touch, reading, and guiding mental imagery exercises. We recommend ICU diaries be used as a tool for nurses, family members, and patients to share emotions, communicate support, and document the patient’s hospitalization, thereby restoring any gaps in memory [133]. More widespread use of this tool is indicated based on previous research supporting its feasibility and perceived benefit in pediatric populations [85,86,87] as well as reductions in anxiety, depression, quality of life concerns, and PMTS in adult ICU patients and their families [134]. We also strongly recommend provision of psychoeducation materials to parents about PICS and how to support child coping. The Society of Critical Care Medicine hosts a resource library at https://www.sccm.org/MyICUCare/Resource-Library (accessed on 1 August 2022), with free educational materials topics spanning ICU hospitalization, discharge, and post-discharge. When feasible, we also recommend encouraging families with young children to engage in child coping strategies and interactive materials. Numerous relatively low effort but intentional TIC practices have the potential to mitigate patient and family distress and risk for PMTS.

If patients or family members have known trauma triggers or reminders, efforts should be made to minimize exposure to them. Environmental modifications include minimizing alarm volume and reducing the presence or visibility of potentially triggering weapons on security guards. Practices that limit one’s autonomy, such as use of physical restraints, should also be used sparingly as they have been associated with patient outcomes including PTSD, delirium, and longer duration of mechanical ventilation [135]. When discussing prognosis, treatment and care options, and potential morbidities, it is essential for providers to be mindful of the cognitive effects of trauma and acute stress including disrupted attention, executive functioning, and memory. Trauma-informed strategies include assessing health literacy, inquiring about and responding to communication preferences, and presenting information in multiple modalities [97].

### 4.4. Integration Recommendations

Full integration of TIC will require significant institutional support, as well as small daily changes in the practices of individual providers. One structural approach is incorporating behavioral health providers into PICU teams. Emerging evidence from two hospitals suggests that such models can support screening, intervention, and referral to needed services. Integrated behavioral health providers may also champion provider wellbeing and reduce the burden on medical providers, given that psychological trauma-specific services (e.g., intervention for acute stress symptoms) are outside of the scope of training for physicians and nurses. Current and future research can help guide efforts to implement TIC in the PICU through use of integrated behavioral health models. Pediatric step down and floor units, as well as post-discharge follow-up clinics would benefit from integrating TIC and increased awareness of post-PICU distress for patients and family members. For children who remain hospitalized after leaving the PICU, care should be taken to minimize PMTS, support coping with current distress (historical and admission-related) and facilitate connection to post-discharge behavioral health services as needed.

## 5. Conclusions

While the TIC approach is beginning to permeate certain aspects of care within the PICU, the evidence suggests that these settings are generally not yet providing comprehensive trauma-informed care. In fact, most providers have not had adequate training in the science of trauma [46,47], recognizing and assessing for traumatic stress reactions [54], nor effectively and appropriately implementing strategies to mitigate traumatic stress and promote resilience [99]. Despite the limited evidence to date, emerging work indicates strong potential for the patient and family experience to be greatly enhanced via TIC.

## Figures and Tables

**Figure 1 children-09-01575-f001:**
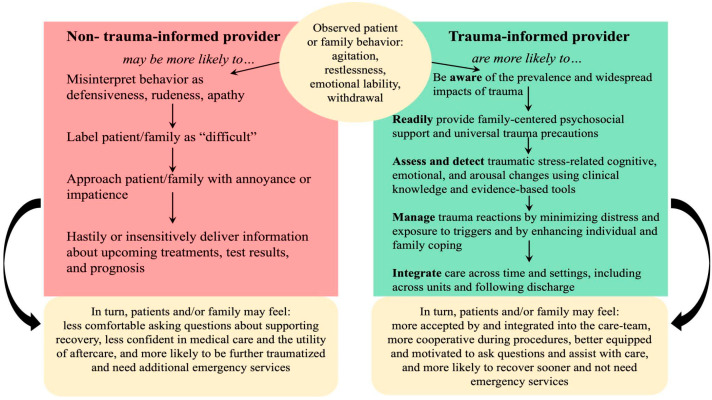
Differential potential reactions and outcomes in response to signs of patient or family traumatic stress.

## Data Availability

Not applicable.

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
