# Peer review of "Is Pediatric Intensive Care Trauma-Informed? A Review of Principles and Evidence"

_children, 2022, doi:10.3390/children9101575_

Round 1

Reviewer 1 Report

Thank you for allowing me to review the manuscript "Is paediatric intensive care trauma-informed? A review of principles and evidence". The authors attempted to review a very important subject that has not received great attention thus far. Reviewing the subject and presenting useful tools to detect, assess and manage psychological trauma and PMTS in paediatric ICU is timely, important and necessary. I have some minor comments:

The review is comprehensive but too long, with several themes repeating throughout the text. It would benefit from shortening and focusing the individual themes and parts of the review. 

Some statements should be moderated or contextualised.

E.g. line 284 stating that families "should be given decision making power"

perhaps a better statement: families should share in decision making

Author Response

We agree that the review was too long and included redundancy in its original form. We appreciate this feedback and have cut over 600 words to minimize repetition throughout the text. We have also made edits to moderate or contextualize statements, such as the one made in line 284. We believe that this editing has greatly improved the manuscript.

Reviewer 2 Report

The manuscript is a comprehensive and a detailed review of  trauma informed care in Pediatric Intensive Care Unit. It is a well written paper and I think that it  can be accepted without any further changes.

Author Response

Thank you for reviewing our manuscript and for your support.